# Stage-specific epigenetic regulation of CD4 expression by coordinated enhancer elements during T cell development

Priya D. Issuree[1], Kenneth Day[2,7], Christy Au[1,3], Ramya Raviram[4], Paul Zappile[5], Jane A. Skok[4], Hai-Hui Xue [6], Richard M. Myers[2] & Dan R. Littman[1,3,4]

The inheritance of gene expression patterns is dependent on epigenetic regulation, but the establishment and maintenance of epigenetic landscapes during T cell differentiation are incompletely understood. Here we show that two stage-specific *Cd4 cis*-elements, the previously characterized enhancer E4p and a novel enhancer E4m, coordinately promote *Cd4* transcription in mature thymic MHC-II-specific T cells, in part through the canonical Wnt pathway. Specifically, E4p licenses E4m to orchestrate DNA demethylation by TET1 and TET3, which in turn poises the *Cd4* locus for transcription in peripheral T cells. *Cd4* locus demethylation is important for subsequent *Cd4* transcription in activated peripheral T cells wherein these *cis*-elements become dispensable. By contrast, in developing thymocytes the loss of TET1/3 does not affect *Cd4* transcription, highlighting an uncoupled event between transcription and epigenetic modifications. Together our findings reveal an important function for thymic *cis*-elements in governing gene expression in the periphery via a heritable epigenetic mechanism.

[1] The Kimmel Center for Biology and Medicine of the Skirball Institute, New York University School of Medicine, New York, NY 10016, USA. [2] Hudson Alpha Institute for Biotechnology, Huntsville, AL, USA. [3] Howard Hughes Medical Institute, New York University School of Medicine, New York, NY 10016, USA. [4] Department of Pathology, New York University School of Medicine, New York, NY 10016, USA. [5] Genome Technology Center, New York University School of Medicine, New York, NY 10003, USA. [6] Department of Microbiology and Immunology, Carver College of Medicine, University of Iowa, Iowa City, IA 52242, USA. [7] Present address: Intermountain Precision Genomics, Translational Science Center, 292 S 1470 E Suite 201, St George, UT 84790, USA. Correspondence and requests for materials should be addressed to D.R.L.(email: dan.littman@med.nyu.edu)

One of the major unsolved problems in developmental immunology is the mechanism by which mature MHC class II (MHC-II)-restricted CD4+ T cells (with helper or regulatory phenotypes) and class I-restricted CD8+ cytotoxic T cells arise from bipotential CD4+8+ (double positive, DP) precursors in the thymus. CD4 and CD8 are co-receptors that are critical to the development of these lineages as they promote TCR binding and signaling upon binding to MHC-II and MHC-I, respectively. Their fine-tuned expression is required for error-free lineage commitment. Following positive selection, DP thymocytes shut off transcription of Cd8 to become CD4+CD8lo intermediates that then give rise to both CD4+ and CD8+ mature single-positive (SP) thymocytes[1,2]. Singer and colleagues have proposed a "kinetic signaling" model, which posits that thymocytes selected by interaction with MHC-II retain signaling at this stage, upregulate ThPOK, and differentiate into CD4 SP cells[3,4], while down-regulation of CD8 in MHC-I-selected cells results in attenuation of signaling accompanied by increased responsiveness to cytokines, e.g. IL-7, allowing for CD8 re-expression and acquisition of cytotoxic T cell properties[5,6]. It remains unclear, however, whether TCR/coreceptor interactions with MHC/peptide result in distinct proximal signals that guide the lineage decisions. Hence, elucidation of the cis-element/s that control expression of Cd4 in MHC-II-specific CD4 SP cells following positive selection could shed some light on how lineage specification is achieved.

Cd4 expression in DP thymocytes is controlled by a cis-acting proximal enhancer, E4p, situated upstream of the Cd4 transcriptional start site (TSS). Germline deletion of the core 432 bp E4p element abrogates CD4 upregulation at the DN4 to DP transition, but a reduced number of MHC-II-specific thymocytes can nevertheless be selected in Cd4E4PΔ/E4PΔ mice, in part due to upregulation of a moderate amount of CD4 during positive selection signaling, suggesting potential regulation by another cis-element in a stage-specific manner[7]. This view is also supported by the finding that E4p deletion in proliferating mature CD4+ T cells has no effect on maintenance of Cd4 expression. In CD8-lineage cells, repression of Cd4 is mediated by a silencer element, S4, present in the first intron. Germline S4 deletion results in ectopic CD4 expression in cytotoxic lineage cells and also in double-negative (DN) thymocytes, indicating that the gene is reversibly repressed during early development[8]. However, following CD8 SP lineage commitment, S4 is no longer required for continued repression of Cd4, consistent with heritable epigenetic silencing[8,9]. Intriguingly, a targeted deletion encompassing S4 and a region 3′ to it, but not deletion of the S4 core alone, resulted in unstable expression of CD4, suggesting the presence of an enhancer element adjacent to the silencer[10].

We previously showed that Cd4 cis-elements have critical roles in regulating DNA methylation patterns during lineage specification[9]. The locus was relatively hypermethylated in DN and DP thymocytes, but MHC-II-restricted cells underwent a marked loss of methylation marks through enzymatic oxidation of methylcytosine early in the course of positive selection[9]. The enzyme(s) responsible for this process was not defined. In contrast, the Cd4 locus in CD8 SP cells remained hypermethylated, and acquired several new methylation marks following positive selection. These changes in methylation status were dependent on the cis-elements that control Cd4 expression in the respective cell types. In the absence of E4p, the Cd4 locus failed to undergo complete demethylation in CD4-lineage cells, while in the absence of S4 the locus became hypomethylated in CD8-lineage cells, with a methylation pattern similar to that in CD4 SP cells. In CD4-lineage cells mutated in E4p, the extent of gene-body methylation was correlated with a gradual loss of CD4 expression upon proliferation in vitro and in vivo[9]. While deficiency of DNA

methyltransferases resulted in loss of Cd4 silencing in proliferating CD8-lineage cells, no similar causal relationship has been demonstrated for DNA demethylation and CD4 expression in CD4-lineage cells.

In this study, we have aimed to further define the endogenous cis-elements that regulate Cd4 expression during development and ascertain their contributions to transcriptional activity and establishment of epigenetic landscapes. We found that a novel enhancer, termed "maturity" enhancer E4m (due to its inferred activity in mature cells[7]), regulates, with E4p, the expression of Cd4 in late-stage MHC-II-specific thymocytes and in mature T cells. This regulation is mediated, in part, through the downstream components of the canonical Wnt signaling pathway. In the absence of E4m and E4p, Cd4 expression was completely abolished in TCRαβ thymocytes. Comparison of the enhancer mutation phenotypes revealed that both the amount and duration of CD4 expression were critical for error-free lineage choice. E4m was required to promote demethylation initiated by E4p in a stage-specific manner, and in its absence Cd4 was incompletely demethylated. Importantly, the function of these cis-elements was correlated with TET1- and TET3-dependent locus demethylation, suggesting that transcriptional activity may be coupled to demethylation activity. However, lack of demethylation in the absence of TET1/3 did not result in a major Cd4 transcriptional defect in the thymus, but led instead to gradual loss of its expression during proliferation of mature T cells, suggesting that thymic demethylation is required for establishment of stable CD4 expression in dividing mature CD4+ T cells. Moreover, induced deletion of E4p in dividing mature T cells deficient for E4m led to retention of substantial CD4 expression, consistent with a role for another E4p-enabled regulatory element that functions in concert with the TET demethylases during thymocyte development. Thus, the enhancers that regulate Cd4 expression perform multiple functions, including not only direct support of transcriptional activity, but also regulation of the gene's methylation state and entrainment of cis-elements that sustain expression after the cells exit the thymus.

## Results

**E4m regulates Cd4 expression in recently selected and mature CD4+ T cells.** We noticed that following positive selection of MHC-II-specific thymocytes, there was gradual upregulation of CD4 (Supplementary Figs. 1 and 2a), consistent with the proposed activity of a late-acting cis-element[10]. To locate such a cis-element, we examined chromatin accessibility in DP, CD4 SP, and CD8 SP thymocytes, using the Immgen ATAC-seq database, and identified a region 3′ to S4 in the first intron of Cd4 that was preferentially accessible in CD4 SP cells and coincided with the segment that had been suggested to harbor enhancer activity based on comparison of intronic deletions[10] (Fig. 1a). We then employed CRISPR-Cas9 technology to delete approximately 700 bp encompassing the accessible region downstream of S4. Deletion of this region had no effect on CD4 expression in pre-selected TCRβloCD24hiCD69− thymocytes, but there was a substantial reduction of CD4 in recently selected CD4+TCRβhiCD24hiCD69+ and mature CD4+TCRβhiCD24loCD69− thymocytes as well as naïve peripheral CD4+ T cells (Fig. 1b, Supplementary Fig. 2a). Reduced surface CD4 staining was accompanied by lower Cd4 mRNA levels (Supplementary Fig. 2b), suggesting that an intronic enhancer, E4m, is required to drive optimal CD4 expression in αβ T cells following positive selection. The reduction in CD4 expression resulted in a decrease in the proportion of CD4+ cells among mature SP thymocytes and peripheral CD4+ T cells (Supplementary Fig. 2c, d) and in decreased ratios of CD4+ to

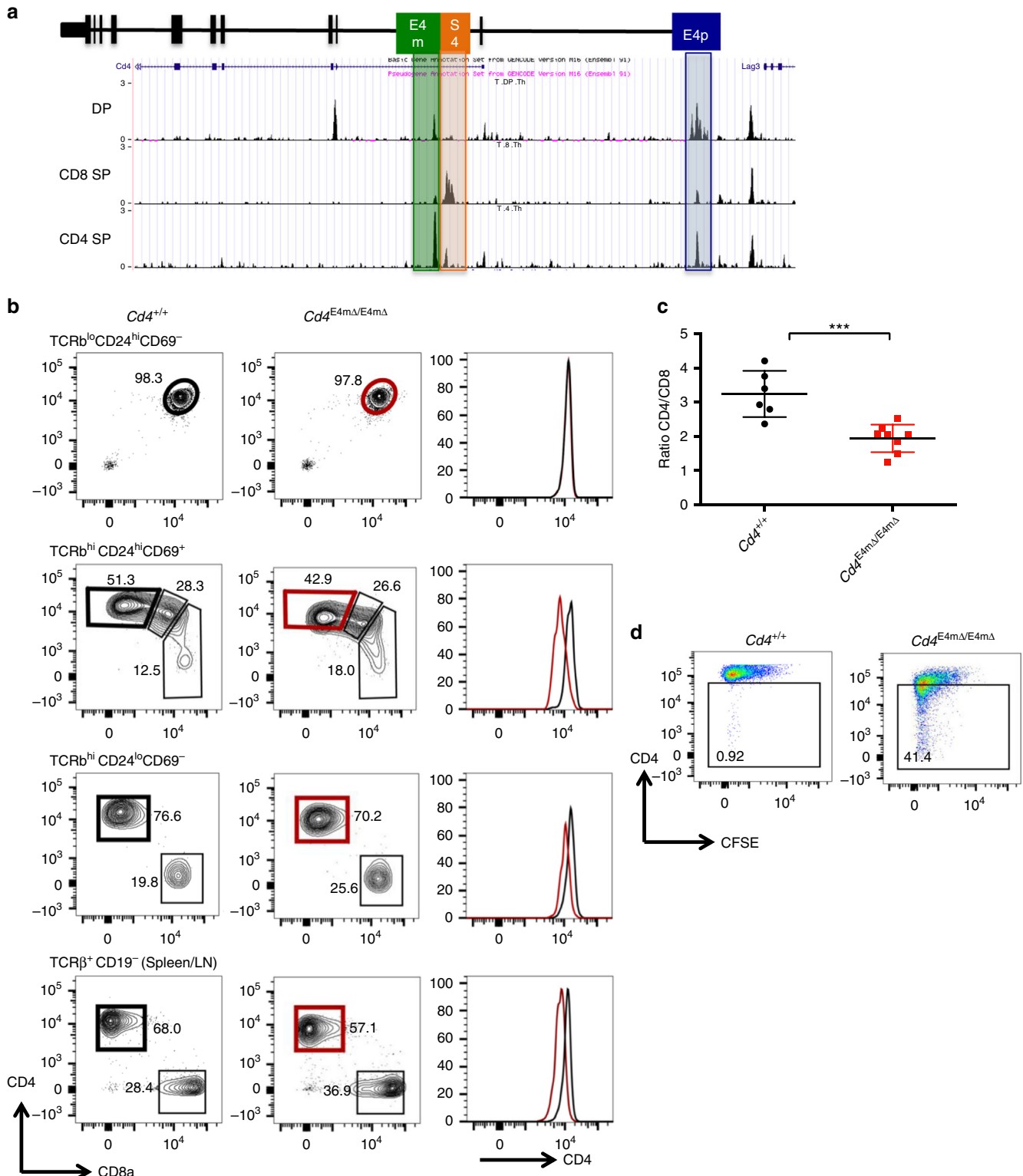

CD8+ T cells in the periphery as compared to wild-type (WT) littermates (Fig. 1c), but we found no statistical difference in the total number of thymocytes between WT and KO mice (Supplementary Fig. 2e). Next, we activated labeled naive CD4+ T cells from *Cd4*^E4mΔ/E4mΔ and control mice in vitro and found a cell-division-coupled loss of CD4 expression in *Cd4*^E4mΔ/E4mΔ cells (Fig. 1d), signifying that E4m is required for the maintenance of stable CD4 expression in activated

peripheral cells in addition to controlling CD4 expression in post-selected CD4+ T cells.

**E4m and E4p are partially redundant in regulating CD4 expression following positive selection.** Although CD4 expression was reduced in the absence of E4m, it was not completely absent, indicating that perhaps E4p still retained enhancer activity

**Fig. 1** E4m regulates *Cd4* expression in recently selected and mature CD4$^+$ T cells. **a** Schematic structure of the *Cd4* locus showing the positions of the *Cd4* proximal enhancer (E4p), the *Cd4* silencer (S4), and the *Cd4* "maturity" enhancer (E4m). Juxtaposed below are genome browser views from ImmGen ATAC-seq datasets from DP, CD8 SP, and CD4 SP thymocytes, respectively. Highlighted boxes show the accessibility of E4m, S4, and E4p. **b** Contour FACS plots showing CD4 and CD8 expression and histograms on the right showing CD4 expression in cell subsets within indicated gates (in black and red) from the thymus and periphery of WT littermates and *Cd4*$^{E4m\Delta/E4m\Delta}$ mice. Numbers in the FACS plot quadrants indicate cell percentages. Representative results of more than three independent analyses. **c** Graph showing the ratio of CD4 to CD8 T cells isolated from the spleen and lymph nodes of WT littermates and *Cd4*$^{E4m\Delta/E4m\Delta}$ mice. Mean + SD. ***$p$ < 0.001(unpaired Student's *t*-test); data are the summary of three independent experiments. **d** FACS plots showing CD4 expression and CFSE dilutions over cell divisions 5 days after in vitro activation with anti-CD3/anti-CD28 of sorted naive CD4$^+$ T cells from mice with indicated genotypes. Numbers in the indicated gate represent cell percentages. Data are representative of more than three independent analyses

in post-selection thymocytes. To test this, we generated mice with germline deletion of both E4m and E4p. In accordance with previous work, there was complete loss of E4p-dependent CD4 expression on pre-selected TCRβ$^{lo}$CD24$^{hi}$CD69$^-$ thymocytes in these mice (Fig. 2a). Strikingly, combined deletion of E4m and E4p resulted in complete absence of CD4-expressing post-selection thymocytes and peripheral T cells (Fig. 2a, b). We also observed a substantial proportion of CD4$^-$CD8$^-$ TCRβ$^{hi}$CD24$^{lo}$CD69$^-$ thymocytes that persisted in the periphery and that were positive for ThPOK expression (Fig. 2c), consistent with their being MHC-class II-restricted CD4 "wannabe" cells, also found in *Cd4*$^{-/-}$ mice[11,12,13]. As we had previously shown that E4p is dispensable once cells migrate to the periphery, we wished to determine if its ability to support expression of CD4 in the absence of E4m was also dependent on the timing of its activity. We therefore generated mice with deletion of E4m and LoxP-flanked E4p (*Cd4* $^{E4m\Delta/E4m\Delta}$ $^{E4pFL/FL}$ mice) and used a retroviral Cre recombinase to excise E4p following activation of mature CD4$^+$ T cells. Remarkably, upon complete excision of E4p in mature T cells deficient for E4m (Supplementary Fig. 3a, b), there was only a partial increased loss of CD4 expression in proliferating cells (Fig. 2d, e). Thus, even though E4p has a role in directing *Cd4* transcription in mature CD4$^+$ T cells, its primary role appears to be to establish heritable transcription of the locus.

**CD4 expression during positive selection is critical to prevent lineage re-direction**. We next tested whether reduction of CD4 expression during T cell development results in errors in lineage choice, with MHC-II-specific thymocytes becoming CD8$^+$ T cells, as previously reported in *Cd4*-deficient mice[14,15]. We bred *Cd4*$^{E4m\Delta/E4m\Delta}$ to MHC-I-deficient (*β2m*$^{-/-}$) mice, in which there is complete loss of CD8 SP cells, and asked whether there were CD8-lineage cells selected on MHC-II. Unlike *Cd4Cre*$^{Tg}$ *Cd4*$^{S4FL/FL}$ mice, which bear a deleted region encompassing both E4m and S4, and in which significant lineage re-direction of MHC-II-selected cells occurs, as previously reported[10], there were fewer than 1% CD8 SP cells in *Cd4*$^{E4m\Delta/E4m\Delta}$; *β2m*$^{-/-}$ mice (Fig. 3a, Supplementary Fig. 4a). In contrast, deletion of E4p led to a substantial proportion of thymocytes redirected towards the CD8 lineage, similar to that in *Cd4Cre*$^{Tg}$ *Cd4*$^{S4FL/FL}$ mice (Fig. 3a). We also assessed lineage re-direction in *Cd4*$^{-/-}$ mice and found a large fraction of CD8 SP cells after bone-marrow transfer of *Cd4*$^{-/-}$ cells into *β2m*$^{-/-}$ hosts (Fig. 3b), confirming that lineage re-direction was a consequence of CD4 expression and not due to enhancer control of another gene in *trans*. We therefore hypothesized that error-free lineage commitment to CD4 SP is dependent on the level of CD4 expression, in addition to the window and duration of CD4 expression. Indeed, comparison of CD4 levels on recently selected TCRβ$^{hi}$CD24$^{hi}$CD69$^+$ thymocytes revealed that CD4 expression in *Cd4Cre*$^{Tg}$ *Cd4*$^{S4FL/FL}$ mice was lower than in *Cd4*$^{E4m\Delta/E4m\Delta}$ mice, and was lowest in *Cd4*$^{E4p\Delta/E4p\Delta}$ mice (Fig. 3c, d; Supplementary Fig. 4b), supporting the idea that a threshold level of CD4 expression is critical

for CD4-lineage commitment. Reduced CD4 expression led to a decrease in Nur77 expression in recently selected CD4$^+$TCRβ$^{hi}$CD24$^{hi}$CD69$^+$ thymocytes from both *Cd4*$^{E4m\Delta/E4m\Delta}$ and *Cd4*$^{E4p\Delta/E4p\Delta}$ mice, consistent with reduced positive selection signaling (Supplementary Fig. 4c, d). However, ThPOK expression was significantly reduced in *Cd4*$^{E4p\Delta/E4p\Delta}$ mice compared to control and *Cd4*$^{E4m\Delta/E4m\Delta}$ mice (Supplementary Fig. 4e), suggesting that continued expression of CD4 was also critical during positive selection for optimal ThPOK upregulation and to allow for proper lineage specification of CD4 SP cells. Interestingly, in contrast to thymocytes, there were few lineage-redirected peripheral CD8$^+$ T cells in *Cd4*$^{E4p\Delta/E4p\Delta}$; *β2m*$^{-/-}$ mice (Fig. 3a). This was not the case in *Cd4*$^{-/-}$; *β2m*$^{-/-}$bone-marrow chimeric mice or *Cd4Cre*$^{Tg}$ *Cd4*$^{S4FL/FL}$ mice (Fig. 3a, b). We therefore tested whether lack of E4p could have an impact on the viability of MHC-Class II-specific thymic CD8 SP cells by performing Caspase-3 staining. We noticed a significant level of Caspase-3$^+$ cells among TCRβ$^{hi}$CD24$^{lo}$ CD8 SP thymocytes of *Cd4*$^{E4p\Delta/E4p\Delta}$; *β2m*$^{-/-}$ mice, but not in control cells (Supplementary Fig. 4f), suggesting that a substantial fraction of MHC-II-selected CD8 SP thymocytes undergo apoptosis, thus explaining the paucity of progeny in the periphery. Concomitantly, we examined IL-7 receptor (IL-7R) expression on CD8 SP cells to assess whether the IL-7/IL-7R signaling axis was involved in providing survival signals to these lineage-redirected cells[16]. We observed a stark decrease in IL-7R expression on TCRβ$^{hi}$CD24$^{lo}$ CD8 SP cells from *Cd4*$^{E4p\Delta/E4p\Delta}$; *β2m*$^{-/-}$ mice compared to control cells (Supplementary Fig. 4g), suggesting that lack of IL-7R signaling on these cells may result in their demise. Taken together these data suggest that both duration and level of CD4 expression are critical for error-free lineage commitment to CD4 SP cells.

**Tcf7/Lef1 control *Cd4* expression in mature CD4$^+$ T cells in a β-catenin-dependent manner**. To identify the transcription factors that drive the expression of CD4 in mature thymocytes, we performed motif analysis of the E4m region and identified binding sites for TCF1 (encoded by *Tcf7*) and LEF1 transcription factors, which contain conserved high-mobility group DNA-binding domains[17,18] (Fig. 4a). Available ChIP-seq data for TCF1 occupancy in thymocytes revealed signals in E4m, S4, and E4p (Fig. 4a, Supplementary Fig. 5a). TCF1 has been shown to drive transcription of *Cd4* in DP thymocytes in a β-catenin-dependent manner[19] and was also shown to mediate *Cd4* repression in CD8 SP T cells via a physical interaction with Runx3[20]. To assess whether TCF1/LEF1 participate in transcription of *Cd4* in mature CD4$^+$ T cells through regulation of E4m, we treated *Ubc-Cre*ER$^{T2}$ *Tcf7/Lef1* $^{FL/FL}$ and control mice with tamoxifen for 4 days and analyzed CD4 expression on peripheral T cells 4 days later. This allowed us to monitor the effect of *Tcf7/Lef1* loss on mature T cells even though *Tcf7/Lef1* deficiency leads to a major loss of thymocytes due to impaired survival[21]. Mice with deletion of *Tcf7/Lef1* had reduced CD4 expression in naive CD4$^+$ T cells

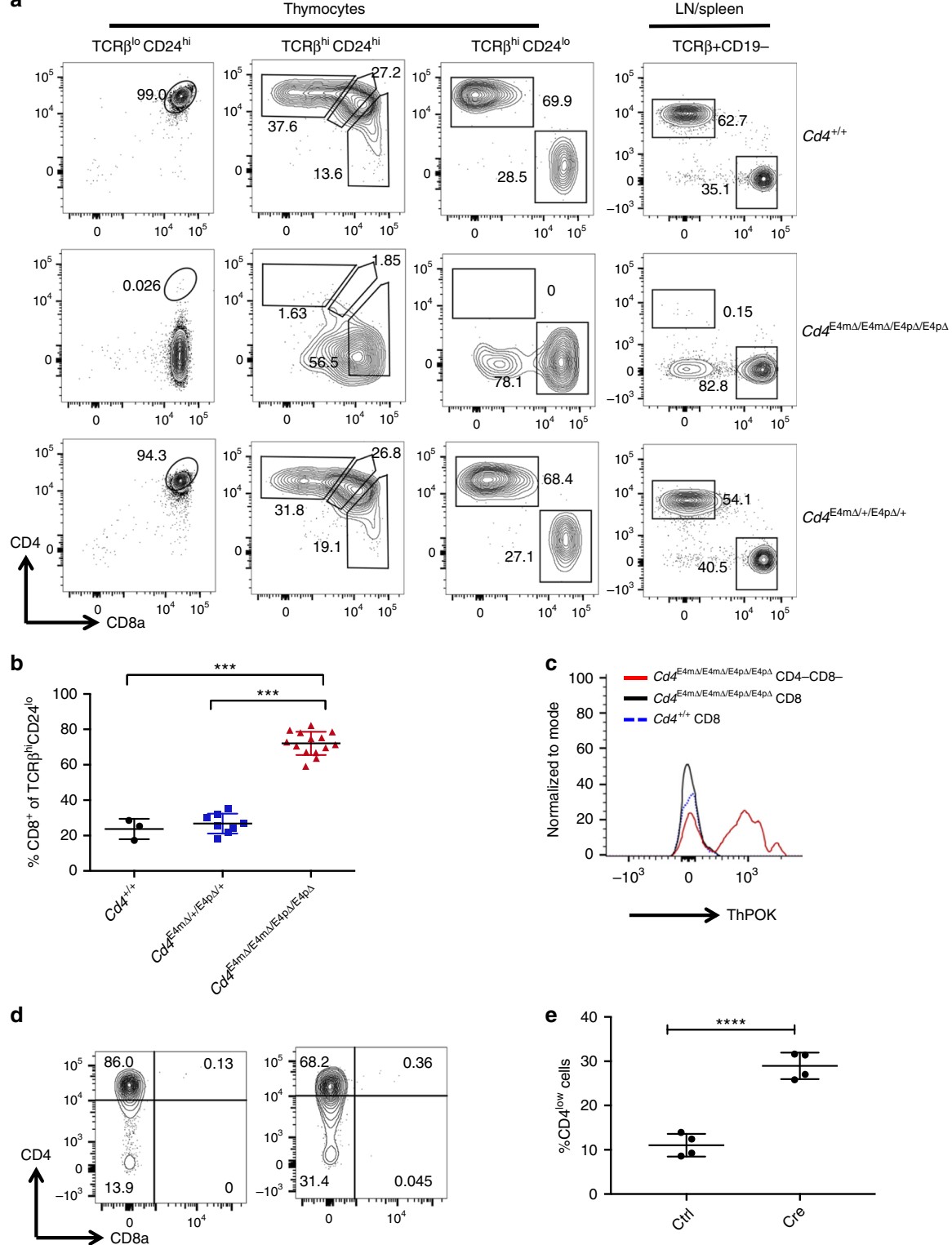

**Fig. 2** E4m and E4p are partially redundant in regulating CD4 expression following positive selection. **a** CD4 and CD8 expression in indicated cell subsets and genotypes. Numbers in the FACS plot quadrants indicate cell percentages. Representative results of more than three independent analyses. **b** Proportions of CD8 SP cells among TCRβ[hi] CD24[lo]CD69[−] thymocytes from mice with the indicated genotypes. Mean + SD. ***p < 0.001 (unpaired Student's t-test). **c** Flow cytometry analysis of ThPOK expression in thymocyte subsets from mice with indicated genotypes. Cells were gated on CD24[lo]CD69[-]TCRβ[hi] prior to further gating based on CD8 expression. Data are representative of two independent experiments with multiple mice. **d** CD4 expression 5 days after in vitro activation and transduction of sorted naive CD4[+] T cells from Cd4 [E4mΔ/E4mΔ E4pFL/FL] mice. Cells were sorted and transduced with retroviral MSCV-Cre-IRES-GFP or control vector. Cells were gated on GFP[+] transduced cells. Data are representative of two independent experiments with technical replicates. **e** Quantification of the percentage of cells that lost CD4 expression (CD4[low]) after activation and transduction with a retroviral MSCV-Cre-IRES-GFP or control vector. Mean + SD. ***p < 0.001(unpaired Student's t-test)

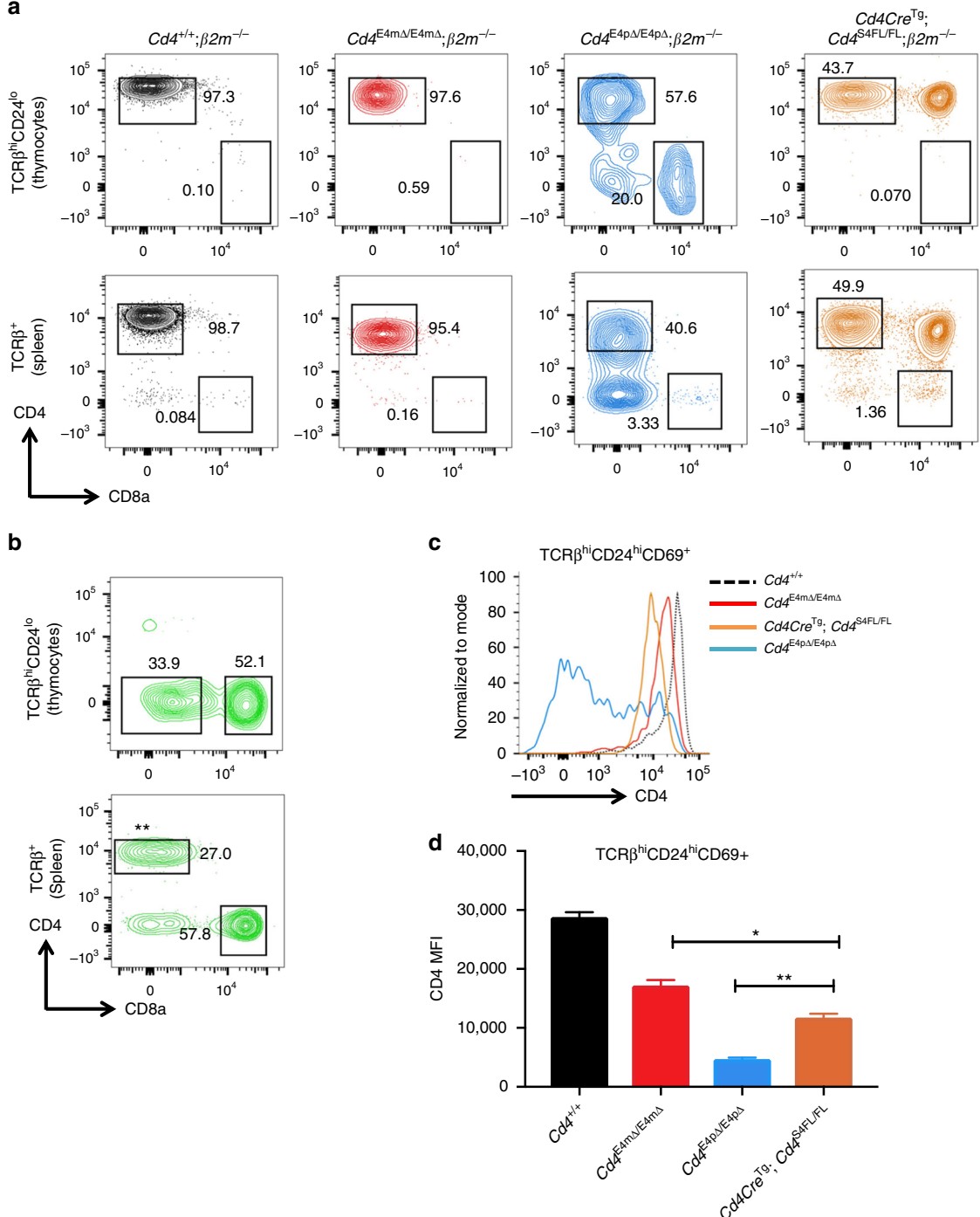

**Fig. 3** *Cd4* expression during positive selection is critical to prevent lineage re-direction. **a** CD4 and CD8 expression in indicated cell subsets and genotypes. Numbers in the FACS plot quadrants indicate cell percentages. Representative results of three independent analyses with multiple mice. **b** CD4 and CD8 expression on indicated cell types, 8 weeks post transfer of $Cd4^{-/-}$ bone marrow into lethally irradiated $\beta2m^{-/-}$ hosts. ** represents un-depleted CD4$^+$ T cells in the periphery of $\beta2m^{-/-}$ hosts. Experiment was performed three times with two or more mice each time. **c** Histogram showing *Cd4* expression on TCRβ$^{hi}$CD24$^{hi}$CD69$^+$ thymocytes of mice with the indicated genotypes. Data are representative of three independent experiments with multiple mice. **d** Graph quantifying the MFI of CD4 on recently selected thymocytes from mice shown in **c**. Data are representative of three independent experiments, with each experiment consisting of two mice in each group. Mean + SD. *$p < 0.05$, **$p < 0.01$ (unpaired Student's *t*-test)

compared to control animals (Fig. 4b, Supplementary Fig. 5b). We also activated naïve CD4$^+$ T cells from control and *Ubc-CreER*$^{T2}$ *Tcf7/Lef1* $^{FL/FL}$ mice in vitro in the presence of 4-hydroxytamoxifen or vehicle control, and found that deletion of *Tcf7/Lef1* in that setting similarly led to decreased CD4 expression (Supplementary Fig. 5c). We then tested whether TCF1/LEF1

regulate *Cd4* expression through the β-catenin pathway, by overexpressing in activated CD4$^+$ T cells a stabilized mutant form of β-catenin lacking the glycogen synthase kinase 3 (GSK) domain[22]. Compared to WT CD4$^+$ T cells transduced with a control vector, we found a significant increase in CD4 expression upon overexpression of β-catenin (Fig. 4c, d). However, this

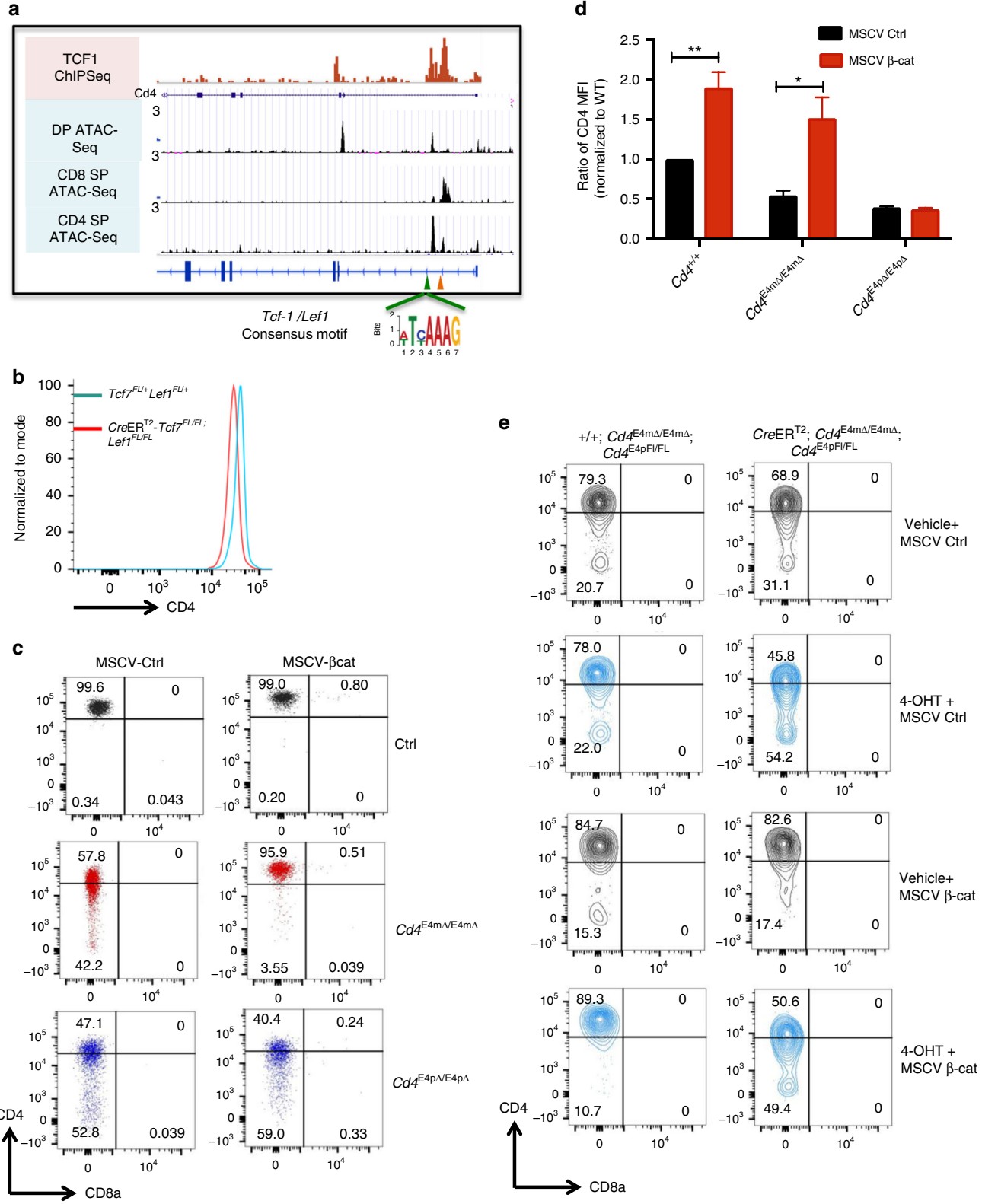

increase was abrogated upon deletion of *Tcf7/Lef1* and less profoundly so when *Tcf7* alone was deleted (Supplementary Fig. 5d), suggesting that both TCF1 and LEF1 drive the expression of *Cd4* in a β-catenin-dependent manner. To determine whether TCF1/LEF1 acted on E4m to drive expression of *Cd4* in mature CD4 T cells, we performed rescue experiments with constitutively active β-catenin in activated *Cd4*$^{E4m\Delta/E4m\Delta}$ or *Cd4*$^{E4p\Delta/E4p\Delta}$ CD4

$^+$ T cells. Interestingly, β-catenin rescued CD4 expression in *Cd4*$^{E4m\Delta/E4m\Delta}$ cells but not in *Cd4*$^{E4p\Delta/E4p\Delta}$ cells, implying that binding of TCF1/LEF1 to E4p is critical for *Cd4* transcription in mature cells (Fig. 4c, d). Similarly, we performed rescue experiments with β-catenin in *Ubc-Cre*ER$^{T2}$*Cd4*$^{E4m\Delta/E4m\Delta}$ E4p$^{FL/FL}$ CD4 T cells, and found that upon inducible deletion of E4p with 4-hydroxytamoxifen there was no enhancement of CD4

**Fig. 4** *Tcf7* and *Lef1* control *Cd4* expression in mature CD4$^+$ T cells in a β-catenin-dependent manner. **a** ChIP-seq dataset of TCF1 in thymocytes (Geo accession GSE46662) juxtaposed onto genome brower views of ImmGen ATAC-seq datasets from DP, CD4 SP, and CD8 SP thymocytes. E4m and S4 are indicated by green and orange arrows, respectively. Consensus TCF1/LEF1 motif sequence found within E4m is highlighted. **b** Histogram showing CD4 expression on sorted TCRβ$^+$CD4$^+$ splenocytes from mice with the indicated genotypes after treatment with tamoxifen. Cells were treated for 4 days, and isolated 4 days after the last treatment. Data are representative of three independent experiments with multiple mice. **c** FACS plots showing CD4 expression on sorted CD4$^+$ T cells from mice with the indicated genotypes after transduction with MSCV IRES-GFP or MSCV-beta-catenin-IRES-GFP vectors. Cells were analyzed for CD4 expression 5 days post transduction. Data are representative of two independent experiments. **d** Graph showing the quantification of CD4 MFI on activated T cells from mice with the indicated genotypes, as shown in the representative FACS plots in **c**. Data are a summary of two experiments done with technical duplicates. MFI values were normalized to MFI of WT cells transduced with the MSCV IRES-GFP control vector. Mean + SD. *$p < 0.05$, **$p < 0.01$ (unpaired Student's $t$-test). **e** CD4 expression on T cells from mice of the indicated genotypes after transduction with MSCV IRES-GFP or MSCV-beta-catenin-IRES-GFP vectors. Naïve CD4$^+$ T cells from the respective mice were FACS sorted and activated in the presence of 4-hydroxytamoxifen or vehicle control for 24 h prior to transduction. Cells were analyzed for CD4 expression 5 days post transduction. Data are representative of two independent experiments done in duplicates

expression (Fig. 4e). To assess whether loss of TCF1/LEF1 during thymic selection results in unstable CD4 expression, similarly to that observed with activated *Cd4*$^{E4pΔ/E4pΔ}$ CD4$^+$ T cells, we activated naïve CD4$^+$ T cells from CD4Cre$^{Tg}$ *Tcf7*$^{FL/FL}$ *Lef1*$^{FL/FL}$ mice, in which there is reduced CD4 expression in peripheral CD4$^+$ T cells comparable to that in *Cd4*$^{E4pΔ/E4pΔ}$ mice (Supplementary Fig. 5e), and noted further loss of CD4 upon cell division (Supplementary Fig. 5f). This loss of CD4 could be partially rescued upon knockdown of DNMT1 (Supplementary Fig. 5g). Interestingly, however, deletion of *Tcf7/Lef1* post-thymically in activated T cells did not lead to unstable CD4 expression, but only to a small decrease in surface level (Supplementary Fig. 5h, i), echoing what was observed upon deletion of E4p in mature T cells. This suggests that E4p, to which the transcription factors TCF1/LEF1 may bind, is required in thymus for the establishment of epigenetic marks that are critical for stable *Cd4* expression in activated T cells[7]. Taken together, these results demonstrate that E4p directs *Cd4* expression at least in part through the TCF1/LEF1-β-catenin pathway in mature CD4$^+$ T cells and that this transcriptional activity is critical early during thymic development to ensure subsequent epigenetically stable expression when cells are activated in the periphery.

**E4m and E4p are both required for the establishment of DNA demethylation necessary for stable CD4 expression.** Following activation of CD4$^+$ T cells from enhancer mutant mice, the gradual cell-division-dependent loss of CD4 expression was more pronounced in *Cd4*$^{E4pΔ/E4pΔ}$ than in *Cd4*$^{E4mΔ/E4mΔ}$ cells (Fig. 5a). We previously reported that, in addition to directing transcription of *Cd4* in DP cells, E4p was required for the initiation of important demethylation activity during positive selection, and that in its absence there was retention of 5mC marks and acquisition of new marks following T cell activation[9]. We therefore assessed whether E4m was also required for demethylation activity at the *Cd4* locus. We isolated genomic DNA from pre-selected TCRβ$^{lo}$CD24$^{hi}$CD69$^-$CD4$^+$CD8$^+$ and mature TCRβ$^{hi}$CD24$^{lo}$CD69$^-$CD4$^+$ thymocytes from WT and *Cd4*$^{E4mΔ/E4mΔ}$ mice, enriched for the *Cd4* locus flanked on each side by ~75 kb using CATCH-Seq, and performed bisulfite sequencing (Fig. 5b, c, Supplementary Fig. 6)[9,23]. The methylation profile at the *Cd4* TSS-proximal region was almost identical in TCRβ$^{hi}$CD24$^{lo}$CD69$^-$CD4$^+$ thymocytes from *Cd4*$^{E4pΔ/E4pΔ}$ and *Cd4*$^{E4mΔ/E4mΔ}$ mice (Fig. 5b). Further, when we stimulated naïve *Cd4*$^{E4mΔ/E4mΔ}$ CD4$^+$ cells to proliferate in vitro and analyzed *Cd4* methylation patterns locus-wide in sorted CD4$^{hi}$ and CD4$^{lo}$ cells after >5 cell divisions, cells that lost CD4 expression exhibited more TSS-proximal differential methylation, similarly to that found in activated *Cd4*$^{E4pΔ/E4pΔ}$ T cells expressing low CD4 (Fig. 5c). Enhanced DNA methylation thus correlates with increased loss of CD4 expression during cell division. To

determine if there may be a causal link between *Cd4* locus gain in methylation and unstable CD4 expression, we interfered with DNMT1 and DNMT3a expression in *Cd4*$^{E4mΔ/E4mΔ}$ CD4$^+$ T cells by using a retroviral shRNA knockdown approach. Knockdown of DNMT1 partially rescued CD4 expression on activated *Cd4*$^{E4mΔ/E4mΔ}$ cells while DNMT3a had a minor effect (Supplementary Fig. 6d, e), suggesting that the absence of E4m during development results in hypermethylation of the *Cd4* locus and hence in reduced expression.

Notably, in contrast to deletion of E4p, E4m deficiency did not affect methylation patterns −9270bp to −15869bp relative to the *Cd4* TSS (Supplementary Fig. 6a) and we did not observe any defect in DNA methylation in DP thymocytes in the absence of E4m compared to WT (Supplementary Fig. 6b), consistent with E4m acting only in post-selection CD4 T cells and the activity of E4m guiding the demethylation process in a stage-specific manner. As the TSS-proximal methylation profiles were similar in cells from *Cd4*$^{E4pΔ/E4pΔ}$ and *Cd4*$^{E4mΔ/E4mΔ}$ mice, and in light of direct or indirect TCF1 occupancy at E4m, as revealed by ChIP-seq, we hypothesized that E4m and E4p are in close proximity to each other, allowing for focused removal of 5mC by TET enzymes. To test this, we performed Capture-C experiments, an adaptation of the conventional 3C approach[24], using capture probes to E4m and S4 in thymocytes at various stages of development and naïve T cells. We detected interaction of E4m with an upstream region that coincided with the location of E4p, but surprisingly found that this interaction was not dynamic and was maintained irrespective of developmental state, even when *Cd4* was repressed (Fig. 5d), in contrast to a previous report[25]. As a control for non-specific E4m site captures, we used naïve CD4$^+$ T cells from *Cd4*$^{E4mΔ/E4mΔ}$ mice and found minimal upstream interaction with E4p, while capture using S4 probes was unaffected and still resulted in upstream interaction with E4p (Fig. 5d, e). Therefore, we conclude that the interaction of E4m with E4p reflects a constitutive structural interaction and suggests that this proximity allows for cooperative TET enzyme-mediated demethylation in the region.

**TET1 and TET3 are required for E4m- and E4p-dependent demethylation of *Cd4*.** To elucidate whether TET enzymes are required for demethylation of the *Cd4* locus, we focused on TET1 and TET3. All TET enzymes are upregulated at the mRNA level in TCRαβ thymocytes during selection (ImmGen RNAseq database), but we examined mutants for TET1 and TET3 because TET2/3 DKOs have a severe defect in T cell differentiation in the thymus and mice develop severe lymphadenopathy and weight loss at 3–5 weeks of age (unpublished observations[26]). There was a decrease of ~10% in the proportion of CD4 SP T cells and naïve CD4$^+$ T cells in the periphery but no statistical differences in total thymocytes and peripheral

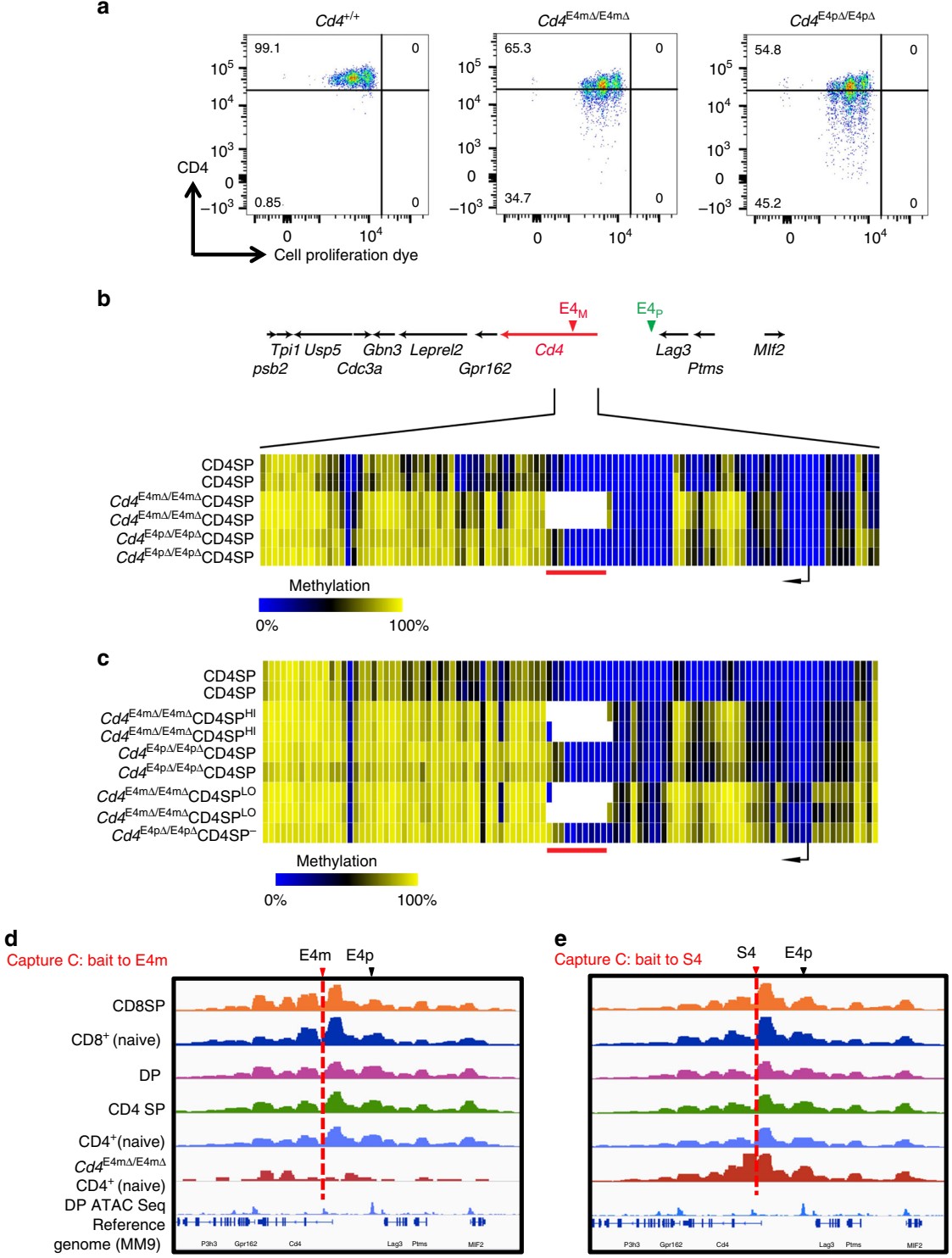

**Fig. 5** E4m and E4p are both required for DNA demethylation necessary for stable *Cd4* expression. **a** CD4 expression as a function of cell proliferation at 72 h after in vitro activation with anti-CD3/anti-CD28 of sorted naive CD4⁺ T cells from mice with the indicated genotypes. Cells were loaded with cell proliferation dye before activation. Numbers in the indicated gates represent cell percentages. Data are representative of more than three independent analyses. **b**, **c** Heatmaps depicting percent CpG methylation in WT CD4 SP, *Cd4*^E4mΔ/E4mΔ^ CD4 SP, and *Cd4*^E4pΔ/E4pΔ^ CD4 SP thymocytes for CpGs from +6200 to −669 relative to the *Cd4* TSS (Chr6:124832027–124838896; mm9). A red line underlines CpGs in E4m (indicated by the gap in the mutant mice) and a black arrow indicates the *Cd4* TSS. CATCH-seq was performed on genomic DNA from sorted populations of TCRβ^hi^CD24^lo^CD69⁻CD4⁺CD8⁻ thymocytes. In **c**, naïve CD4⁺ T cells from mice with the indicated genotypes were sorted, labeled with cell proliferation dye, and stimulated in vitro with anti-CD3/anti-CD28, and cells were sorted 5 days later based on their CD4 expression (high vs. low) for CATCH seq. Replicates were derived from two independent experiments. Note data for *Cd4*^E4pΔ/E4pΔ^ conditions were from previously published experiments with similar experimental conditions[9]. **d**, **e** Overlaid normalized mean Capture-C profiles from indicated cell types with MM9 ATAC-seq dataset from DP thymocytes. The captured E4m and S4 viewpoint fragments are highlighted with the dotted red lines and the interactions with E4p are indicated with black arrows. Data were normalized using DESeq2 using 2 kb sliding window (slide by 100 bps)

T cell numbers in TET1/3 DKO mice compared to littermates (Supplementary Fig. 7a–c). There was also no significant difference in CD4 protein or *Cd4* mRNA expression in recently selected and mature TET1/3 DKO CD4 SP thymocytes compared to littermate controls (Fig. 6a; Supplementary Fig. 7d, e), suggesting that the lack of demethylation does not affect CD4 expression levels during T cell differentiation in the thymus. We noticed a slight decrease in CD4 protein levels and mRNA levels in peripheral naïve CD4$^+$ T cells from TET1/3 DKO mice as well as a decrease in CD8 MFI (Supplementary Fig. 7f–h). In contrast to the findings in thymocytes and naïve CD4$^+$ T cells, there was progressive loss of CD4 expression at protein and mRNA levels of in vitro-activated naïve CD4$^+$ T cells from TET1/3 DKO mice (Fig. 6b; Supplementary Fig. 7i), resembling that observed in activated *Cd4*$^{E4m\Delta/E4m\Delta}$ and *Cd4*$^{E4p\Delta/E4p\Delta}$ T cells and also in proliferating CD4$^+$ T cells deprived of TCF1 and LEF1. We also examined *Thpok*, *Bcl11b*, and *Satb1* mRNA expression in activated TET1/3 DKO CD4 T cells with reduced CD4 expression and found no significant differences (Supplementary Fig. 7j–l), suggesting that loss of *Cd4* was less likely due to *trans* effects of transcription factors that control *Cd4* expression.

Methylation profiling of the *Cd4* locus in TET1/3 DKO mice by CATCH-seq revealed persistence of methylation marks in the gene body of *Cd4* in activated CD4$^+$ T cells from TET1/3 DKO, similarly to activated T cells from *Cd4*$^{E4m\Delta/E4m\Delta}$ mice, while other 5mC marks were lost even in the absence of TET1/3 (Fig. 6c). These results indicate that some methylation marks could be maintained and were not passively diluted out after cell division. Furthermore, the high similarity in methylation patterns in TET1/3 DKO cells compared to *Cd4*$^{E4m\Delta/E4m\Delta}$ and *Cd4*$^{E4p\Delta/E4p\Delta}$ T cells suggests that E4m and E4p may work in a cooperative manner to mediate the recruitment of TET1/3 to the locus. Taken together, we surmise that TET1/3 mediate *Cd4* locus demethylation during T cell maturation in the thymus through the coordinated efforts of E4m and E4p, and that removal of 5mC is required for subsequent maintenance of *Cd4* transcription in the periphery and during T cell activation.

## Discussion

The *cis*-element/s and transcription factors that allow for continued expression of *Cd4* in MHC-II-selected CD4$^+$ T cells following positive selection have not been fully defined. It also remains unclear whether gene body methylation changes are required for appropriate transcription of *Cd4* in CD4 SP T cells during maturation in the thymus or whether methylation changes are a result of transcriptional activity mediated by *cis*-elements and have subsequent critical functions related to heritable transmission of gene states. We have shown here that a second enhancer, E4m, is required together with a previously characterized enhancer, E4p, to drive optimal *Cd4* expression during positive selection, ensure error-free specification of the CD4 lineage, and maintain stable expression in proliferating mature CD4$^+$ T cells. The transcription factors TCF1 and LEF1 are required for optimal E4p activity during development, and enhancer function also dictates DNA demethylation activity throughout the locus. We found that TET1 and TET3 mediate locus demethylation of *Cd4*, likely through the activity of E4m and E4p during thymocyte maturation, but that methylation changes do not impede transcription in non-dividing cells and are instead critical for establishing heritable expression states in dividing cells.

We previously found that DP cells in the thymus have high CD4 expression despite *Cd4* locus hypermethylation comparable to that in cytotoxic CD8$^+$ and DN cells[9], and raised the question

as to whether methylation had a role in *Cd4* repression or was uncoupled from expression of *Cd4*. By utilizing conditional knockouts of TET enzymes, we now demonstrate that, indeed, *Cd4* transcription can be uncoupled from changes in methylation, as there was no defect in expression in CD4 SP cells in the thymus despite hypermethylation, but there was a pronounced effect in peripheral, particularly, in activated proliferating CD4$^+$ T cells. It is plausible that presence of these methylation marks in TET1/3 mutant mice has no direct impact in *cis* on *Cd4* transcription, but affects the expression of other factors required in *trans* for *Cd4* transcription in the periphery and/or during activation. However, given the similar phenotypic features of *Cd4*$^{E4m\Delta/E4m\Delta}$ and *Cd4*$^{E4p\Delta/E4p\Delta}$ T cells in terms of unstable CD4 expression after activation, it is more likely that methylation is acting *in cis*. We therefore postulate that presence of gene body methylation leads to reduced transcription during cell division. TET-mediated demethylation may be required to maintain CD4 levels in circumstances in which E4p and E4m are no longer required in the periphery, particularly if another enhancer takes over the transcriptional function following activation of CD4$^+$ T cells. This possibility is supported by the observations that deletion of E4p and E4m in activated T cells leads to a substantial decrease, but not loss, of CD4 expression, and that deletion of a gene segment containing E4m and S4 in activated CD4$^+$ T cells does not affect CD4 expression[10]. E4p, and possibly E4m, activity in the thymus may thus be required to epigenetically prime or poise another putative enhancer for activity in peripherally activated cells, as germline deletion of both E4m and E4p completely abolished CD4 expression. Alternatively, DNA demethylation could have a heritable role in ensuring higher order chromatin architecture in defined neighborhoods in mature CD4$^+$ cells after activation. Therefore, elucidating how CD4 levels are maintained in the periphery after maturation in the thymus will be key to dissecting the importance of demethylation during CD4$^+$ T cell selection.

Our results using endogenous deletions of *Cd4 cis* elements also have implications for the kinetic signaling model. In accordance with the model, our data demonstrate that CD4 expression is key to propel CD4-lineage specification by ensuring optimal TCR signaling in MHC-II-restricted CD4$^+$CD8$^{lo}$ intermediates. Reduced CD4 expression during selection in *Cd4*$^{E4p\Delta/E4p\Delta}$ and *Cd4Cre*$^{Tg}$ *Cd4*$^{S4FL/FL}$ mice both led to errors in lineage specification. Interestingly, however, *Cd4Cre*$^{Tg}$ *Cd4*$^{S4FL/FL}$ mice had more MHC-II-specific CD8 cells than *Cd4*$^{E4m\Delta/E4m\Delta}$ mice and correlated with a greater defect in CD4 surface expression in *Cd4Cre*$^{Tg}$ *Cd4*$^{S4FL/FL}$ mice. Since we could detect proximal interactions of E4m with E4p by Capture C, it is possible that deletion of a longer gene segment in *Cd4Cre*$^{Tg}$ *Cd4*$^{S4FL/FL}$ mice impacted the function of E4p in the course of positive selection, leading to a pronounced defect in *Cd4* transcription, or that the intervening region between E4m and S4 has a role in modulating *Cd4* transcription. Moreover, we found that MHC-II-specific *Cd4*$^{E4p\Delta/E4p\Delta}$ CD8$^+$ T cells were unable to upregulate IL-7R and survive after selection in the thymus. It is unclear if this defect is due to additional gene targets of the E4p enhancer or a consequence of low CD4 expression at a critical time during selection.

Questions raised by our study include how the enhancer elements direct demethylation in the CD4 lineage and how this process differs from that in CD8$^+$ T cells, in which there may be a requirement for de novo methylation activity during positive selection. Our data allude to a dynamic interplay between DNA methyltransferases and TET enzymes, as illustrated by the gain of methylation in the absence of TET enzymes during activation, and therefore it will be interesting to dissect out the antagonistic mechanisms involved. Meanwhile our data revealed that enhancers are required for demethylation, but it is unclear whether

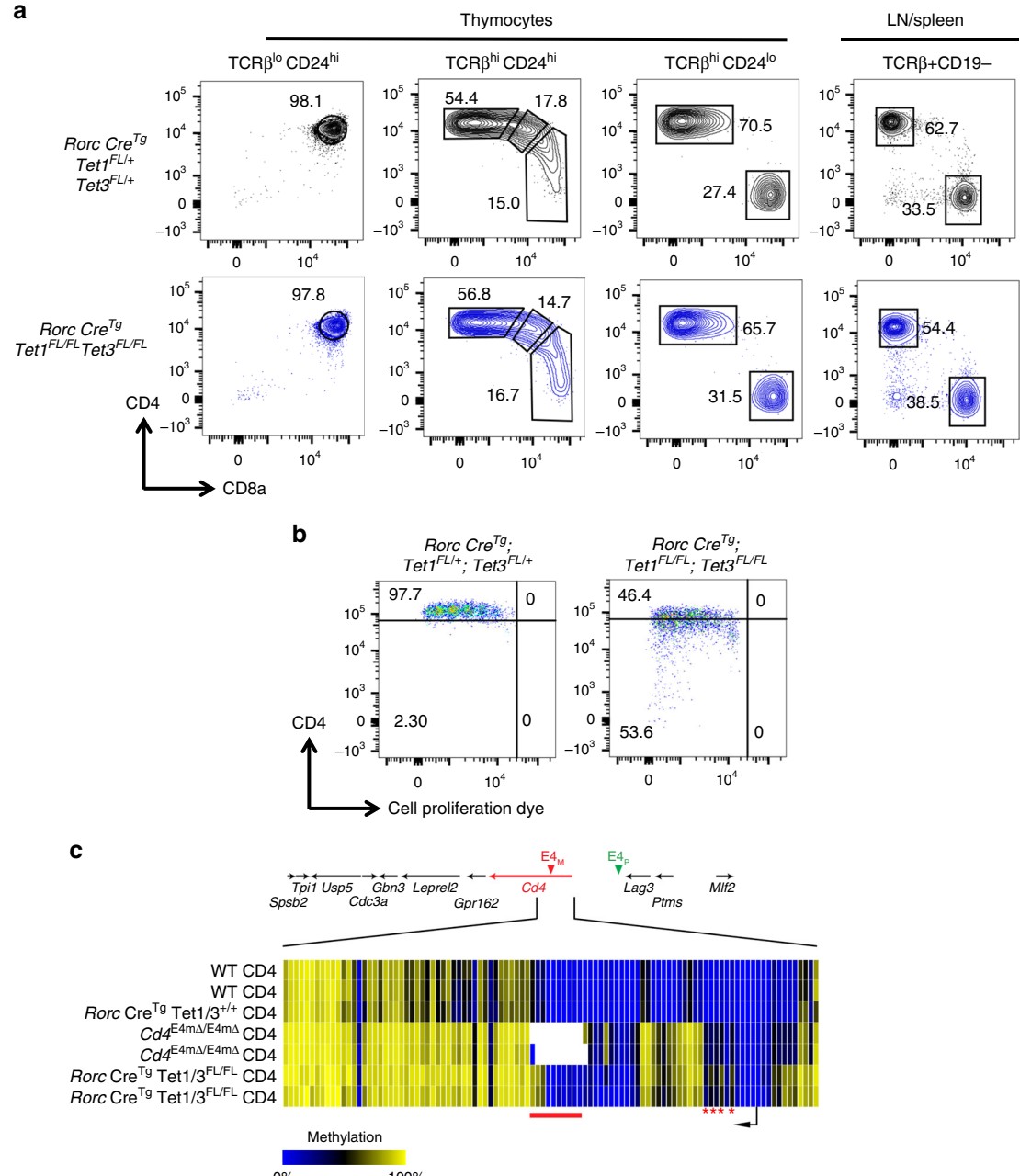

**Fig. 6** TET1 and TET3 are required for E4m- and E4p-dependent demethylation of *Cd4*. **a** FACS plots showing CD4 and CD8 expression in indicated cell subsets from *RorcCre*[Tg] *Tet1*[FL/FL] *Tet3*[FL/FL] and control littermate mice. Numbers in the FACS plot quadrants indicate cell percentages. Representative results of three independent analyses. **b** CD4 expression as a function of cell proliferation, measured by cell proliferation dye dilution, at 3 days after in vitro anti-CD3/anti-CD28 activation of sorted naive CD4$^+$ T cells from mice with indicated genotypes. Numbers in the indicated gates represent cell percentages. Data are representative of more than three independent analyses. **c** Heatmaps depicting percent CpG methylation of residues within the region from +6200 to −669 relative to the *Cd4* TSS (Chr6:124832027–124838896; mm9) in sorted activated CD4$^+$ T cells 5 days post-activation, gating on CD4$^{mid-hi}$ cells for Tet1/3 conditions and CD4$^{hi}$ for *Cd4*$^{E4mΔ/E4mΔ}$ conditions. A red line underlines CpGs in E4m (indicated by the gaps in the E4m mutant mice), and a black arrow indicates the *Cd4* TSS. CATCH-seq was performed on genomic DNA from sorted populations. Red asterisks indicate 5mC marks that are lost in all conditions after cell division

there is recruitment of TET1/3 by specific transcription factors such as TCF1/LEF1 or whether recruitment and activity of TET1/3 are secondary to transcriptional states driven by TCF1/LEF1 and other transcription factors binding to E4p and E4m. Transcriptional activity may result in changes in the local chromatin environment, altering chromatin accessibility and activating histone modifications such as H3K4me3, which may promote local TET function[27,28]. Indeed, the chromatin region around E4m

appears more accessible in CD4 SP than DP cells. Therefore, probing for TET1/3 protein interactors in the thymus and characterization of additional candidate *cis*-acting elements will be critical to advance our understanding of *Cd4* transcriptional regulation at different stages in thymus and beyond. Moreover, further studies of *Cd4* regulation will likely provide insight into how heritable states of gene expression, both positive and negative, are established at distinct stages of development.

## Methods

**Generation of *Cd4* enhancer mutant mice**. Mouse strains carrying deleted enhancer alleles were created using published CRISPR/Cas9 protocols[29,30] at the NYU School of Medicine's Rodent Genetic Engineering Laboratory. Briefly, sgRNAs targeting the DNaseSeq-accessible E4m enhancer region were designed using the Crispr guide design software (http://crispr.mit.edu/) and sgRNA recognition sequences used are listed in the primers and oligonucleotide sequence table. sgRNA-encoding sequences were cloned into a PX458 vector (a gift from Dr. Feng Zhang, Addgene #48138), using oligos with a T7 promoter containing the sgRNA template that were chemically synthetized by IDT. The PCR-amplified T7-sgRNA products were then used as templates for in vitro transcription using the MEGAshortscript T7 kit (Thermo Fisher Scientific) and the mRNAs were purified using the MEGAClear Transcription Clean-up kit (Thermo Fisher Scientific). Mammalian optimized Cas9 mRNA was purchased commercially from TriLink Biotechnologies. Transgenic knock-in mice were generated by injection of a mixture of Cas9 mRNA (100 ng/μl) and sgRNA (50 ng/μl each guide) in injection buffer (10 mM Tris, pH 7.5; 0.1 mM EDTA) into the cytoplasm of C57BL/6J embryos in accordance with standard procedures approved by the IACUC at the NYU School of Medicine. Female CD-1 mice (Charles River) were used as foster mothers. F0 mice were genotyped by PCR amplication of the deleted region and TA cloning, followed by Sanger sequencing. Founders bearing E4m deletions in chromosome 6 with MM9 coordinates 124 835 018–124 835 714 were then backcrossed at least one time to wild-type C57BL/6J mice. For the generation of $Cd4^{E4mΔ/E4mΔ}$ $E4pΔ/E4pΔ$ and $Cd4^{E4mΔ/E4mΔ}$ $E4pFL/FL$ mice, $Cd4^{E4pΔ/E4pΔ}$ and $Cd4^{E4pFL/FL}$ embryos were used, respectively, for CRISPR injections. Founders bearing deletions mapping to MM9 coordinates 124 835 012–124 835 635 and 124 834 997–124 835 709 were then bred to establish homozygous lines.

**Mice**. $Cd4^{E4pΔ/E4pΔ}$.[7], $Cd4^{E4pFL/FL}$[7], $Cd4Cre^{Tg}Cd4^{S4FL/FL}$[8], $Cd4Cre^{Tg}$ $Tcf7^{FL/FL}Lef1^{FL/FL}$ and $UbcCreER^{T2}Tcf7^{FL/FL}Lef1^{FL/FL}$[20] strains were previously described. $Cd4^{-/-}$ ($Cd4^{tm1Mak}$) and $B2m^{-/-}$ ($B2m^{tm1Unc}$), $UbcCreER^{T2}$ (B6.Cg-$Ndor1^{Tg(UBC-cre/ERT2)1Ejb}$/1J) mice were purchased from the Jackson Laboratory. $Tet1$ floxed (C57BL/6) and $Tet3$ floxed mice (129 backcrossed to C57BL/6) were kindly provided by Dr. Iannis Aifantis and Dr. Yi Zhang, respectively. These mice were then backcrossed onto $RorcCre^{Tg}$ (B6.FVB-Tg(Rorc-cre)1Litt/J) mice. All mice were maintained under specific pathogen-free conditions in the Skirball Institute Animal Facility. All experiments were performed in accordance with the protocol approved by the IACUC at the NYU School of Medicine.

**T cell activation and retroviral transduction**. Tissue culture plates were coated with polyclonal goat anti-hamster IgG (MP Biomedicals) at 37 °C for at least 2 h and washed 3× with PBS before cell plating. FACS-sorted CD4⁺CD8⁻CD25⁻CD62L⁺CD44lo naïve T cells were seeded in T cell medium, along with anti-CD3 (BioXcell, clone 145-2C11, 0.25 μg/ml) and anti-CD28 (BioXcell, clone 37.5.1, 1 μg/ml) antibodies. Forty-eight hours later cells were lifted off the plates and cultures were supplemented with 100 U/ml recombinant human IL-2 (Peprotech). For 4-hydroxytamoxifen (4-OHT) treatment, cells were incubated with 0.25 μM 4-OHT (Sigma-Aldrich) 16 h after seeding with anti-CD3/anti-CD28.

For *Dnmt1* and *Dnmt3a* knockdowns, shRNA plasmids on a mIR-E backbone were provided by Johannes Zuber[31]; MSCV-beta-catenin-IRES-GFP was a gift from Tannishtha Reya (Addgene plasmid #14717). MSCV-Cre-IRES-GFP was previously described[7]. Retroviruses were packaged in PlatE cells by transient transfection using TransIT 293 (Mirus Bio). Cells were transduced by spin infection at $1200 \times g$ at 32 °C for 90 min in the presence of 10 μg/ml polybrene (Sigma) 12–16 h postactivation with anti-CD3/anti-CD28. Viral supernatants were removed the next day and replaced with fresh medium containing anti-CD3/anti-CD28. Cells were lifted off 24 h later and supplemented with IL-2.

**Methylation analysis**. Genomic DNA was isolated from FACS-sorted T cell populations using genomic DNA isolation kits (Qiagen). For locus-wide bisulfite sequencing, CATCH-seq was performed as previously described[23] using BAC clone RP24-330J12 (BACPAC Resource Center, CHORI).

**Flow cytometry and sorting**. Monoclonal antibodies were purchased from Thermo Fisher Scientific eBioscience or BD Bioscience. Clones used were: anti-CD4 (RM4-5, dilution 1:500), anti-CD8α (53–6.7 dilution 1:500), anti-TCRβ (H57-597, dilution 1:250), anti-HSA (M1/69, dilution 1:500), anti-CD69 (H1.2F3, dilution 1:250), anti-CD44 (IM7, dilution 1:500), anti-CD62L (MEL-14, dilution 1:500), anti-CD19 (ID3, dilution 1:300), anti-CD127 (SB/199, dilution 1:250), anti-Caspase-3 (C92-605, 10 μl of Ab per test). Cell proliferation dye E450 was obtained from Thermo Fisher Scientific eBioscience. After staining with antibodies and DAPI (Thermo Fisher Scientific), cells were analyzed with an LSRII flow cytometer (BD Biosciences) or sorted with an Aria II (BD Biosciences). Post sort sample purity was >98%. In most cases, anti-CD4, anti-CD8, anti-PE, and anti-B220 magnetic beads (Miltenyi) were used for enrichment and depletion on MACS columns (Miltenyi) before sorting. Flow cytometry data were analyzed using Flowjo software (Tree Star).

**Bone-marrow chimeras**. 4- to 6-week-old C57BL/6 $β2m^{+/+}$ or $β2m^{-/-}$ mice were used as recipients and lethally irradiated 24 h before bone-marrow transfer. Bone marrow from 5- to 6-week-old $Cd4^{-/-}$ donor mice were then harvested and T cells were depleted using anti-TCR-β PE antibody and anti-PE (Miltenyi) magnetic beads. After retro-orbital transfer of bone marrow, mice were kept on antibiotics for 2 weeks and were euthanized for analysis 6–8 weeks post transfer.

**Real-time quantitative PCR**. DNase I-treated total RNA was prepared from sorted cells using Trizol (Ambion) and RNeasy MinElute Clean-up kits (Qiagen) and cDNA was synthesized by SuperScript IV reverse transcriptase (Thermo Fisher Scientific) using OligodT primers. Quantitative PCR was performed using Lightcycler 480 SYBR Green Master Mix (Roche) and a LightCycler 480 (Roche).

**Capture C sample preparation**. 3C libraries were prepared from sorted populations from the thymus and spleen/lymph nodes as described previously[24] with the following changes: *DpnII* enzymes were used for restriction digest, with a total of 500 U fresh restriction enzyme added every 5–7 h, for a total incubation of 16–24 h after initial addition of restriction enzyme. After digest, *DpnII* was heat inactivated and a ligation reaction with 200 U of T4 DNA ligase (NEB) was set up overnight at room temperature with gentle rotation. Two aliquots of 5 μg of 3C library was then sonicated to 200 bp using a Covaris LE220 instrument (190 s shearing time; 30% duty factor; 450PIP, 200 cycles per burst). We indexed libraries with Illumina Truseq indexed sequencing adaptors using NEBnext reagents (E6000, E6040, E7335, or E7500) for end repair, dA labeling, adaptor ligation, and PCR indexing, primarily following the manufacturer's instructions. To maximize library complexity, we used 10 μg of input material split into two reactions and pooled them after indexing. We assessed libraries using an Agilent Bioanalyser or D1000 Tapestation after addition of sequencing adaptors.

**Capture C Oligo design and capture**. Single 120 bp biotinylated DNA oligonucleotides (IDT), which include the *DpnII* restriction sites, were used to capture each end of the target restriction fragment. Primers were blasted across the *Mus musculus* genome to ensure specific binding to the desired regions. The MM9 coordinates of the *DpnII* digested fragments encompassing E4m were 124 835 092–124 835 618; MM9 coordinates of *DpnII* digested fragments containing S4 were 124 835 899–124 836 720.

About 1.5–2 μg of adapter ligated library material was used in a 1.5 ml microcentrifuge tube together with 5 μg human COT DNA (Roche-Nimblegen), 1000 pmol IDT XGen Universal blocking oligo P5 and 1000 pmol IDT XGen Universal Index blocking oligo P7 (corresponding to the Illumina TS index used). The contents of the tube were dried using a vacuum centrifuge (50–60 °C). We resuspended the dried DNA in 7.5 μl Nimblegen 2× Hybridization Buffer and 3 μl Nimblegen Hybridization Component A and then denatured this mix at 95 °C for 10 min. Concurrently, we heated 4.5 μl of the mixed biotinylated capture oligonucleotide library (total 13 pmol) in a 0.2 ml PCR tube to 47 °C in a PCR thermocycler. After denaturation we added the 3C library and blocking oligonucleotides to the biotinylated oligonucleotides without removing them from the heating block and incubated this hybridization reaction in the thermocycler at 47 °C for 64–72 h (with a heated lid at 57 °C). After 2 days, washed Dynabeads (M-270 Streptavidin, Thermo Fisher Invitrogen) were used according to the protocol outlined in Chapter 6 of the SeqCap EZ Library SR User Guide version 5.1 to select for captured DNA.18 cycles of post capture PCR were performed using the NEB PCR master mix and NEB primers. Enrichment of targets before and after a single oligocapture step was verified by qPCR. Libraries were quantitated by qPCR with the KAPA Universal library quantitation kit (Kapa Biosystems) and library size was verified with the Agilent tapestation system on high sensitivity DNA tape. Each capture pool was loaded onto one paired end 150 lane of Illumina Hiseq 4000.

**Capture C sample analysis**. Analysis was performed using the Capture C analysis pipeline from the Hughes lab (https://github.com/Hughes-Genome-Group/captureC). Normalized bedGraph files were generated using DESEq2[32] for 2 kb windows sliding by 100 bps.

**Quantification and statistical analysis**. All *p*-values were calculated using unpaired two-tailed Student's *t*-test with GraphPad Prism7 software, unless otherwise described in the Methods or figure legends. Statistically significant differences are indicated with asterisks in figures with the accompanying *p*-value in the legend. Error bars in figures indicate standard deviation (SD) or standard error of the mean (SEM) for the number of replicates, as indicated in the figure legend.

## Data availability

CATCH-seq and Capture datasets in this study are available under BioProject ID PRJNA481129. Other data that support the findings of this study are available from the corresponding authors upon reasonable request.

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

## Acknowledgements

We thank Dr. Sangyong Kim (NYU RGE core) for help with generation of the various mouse strains. We thank Y. Zhang (Boston's Children Hospital) for Tet3 floxed mice and Iannis Aifantis (NYU School of Medicine) for Tet1 floxed mice. We are grateful to Angela Santana for technical advice and assistance with experiments. We thank Emily Miraldi, Ren Yi, and Richard Bonneau (NYU/ Simmons Foundation) for bioinformatics support to analyze ATAC-seq datasets. Supported by the Cancer Research Institute (to P. D.I.) and the Howard Hughes Medical Institute (to D.R.L.).

## Author contributions

P.D.I. designed and performed experiments with C.A, analyzed data, and wrote the manuscript. K.D. did methylation captures and analyses of *Cd4* locus-wide methylation with support from R.M.M.; R.R. performed the bioinformatics analysis for CaptureC with support from J.A.S. P.Z. performed CaptureC Oligo captures and sequencing. H.-H.X provided the data for in vivo deletion of *Tcf7/Lef1* in peripheral cells as well as mice and reagents. D.R.L. supervised experiments and wrote the manuscript with input from the other authors.

## Additional information

**Competing interests:** The authors declare no competing interests.

