## [Peer Review File · Nature Communications]

REVIEWERS' COMMENTS:

Reviewer #1 (Remarks to the Author):

Authors have addressed my comments. Recommend publication.

Reviewer #2 (Remarks to the Author):

The authors have added some new data that help clarify certain points, although they prefer to keep their data on whole genome sequencing and Vitamin C for a future study. The paper is nevertheless interesting even in its current minimally revised form, and provides novel insights into the epigenetic regulation of CD4 expression.

REVIEWERS' COMMENTS:

Reviewer #1 (Remarks to the Author):

Authors have addressed my comments. Recommend publication.

We are glad to have addressed the reviewer's comments and are thankful for the insights during review.

Reviewer #2 (Remarks to the Author):

The authors have added some new data that help clarify certain points, although they prefer to keep their data on whole genome sequencing and Vitamin C for a future study. The paper is nevertheless interesting even in its current minimally revised form, and provides novel insights into the epigenetic regulation of CD4 expression.

We are glad that the reviewer finds our paper interesting and are thankful for the insights that helped make our manuscript better.